# The Eye as a Diagnostic Tool for Alzheimer’s Disease

**DOI:** 10.3390/life13030726

**Published:** 2023-03-08

**Authors:** Ahsan Hussain, Zahra Sheikh, Manju Subramanian

**Affiliations:** Department of Ophthalmology, Boston University Chobanian & Avedisian School of Medicine, Boston, MA 02118, USA

**Keywords:** Alzheimer’s disease, dementia, ocular biomarkers

## Abstract

Alzheimer’s disease (AD) is a progressive neurodegenerative disorder impacting cognition, function, and behavior in the elderly population. While there are currently no disease-modifying agents capable of curing AD, early diagnosis and management in the preclinical stage can significantly improve patient morbidity and life expectancy. Currently, the diagnosis of Alzheimer’s disease is a clinical one, often supplemented by invasive and expensive biomarker testing. Over the last decade, significant advancements have been made in our understanding of AD and the role of ocular tissue as a potential biomarker. Ocular biomarkers hold the potential to provide noninvasive and easily accessible diagnostic and monitoring capabilities. This review summarizes current research for detecting biomarkers of Alzheimer’s disease in ocular tissue.

## 1. Introduction

Alzheimer’s disease (AD) is a progressive neurodegenerative disorder affecting millions of people globally. AD is currently the most common cause of cognitive impairment among the elderly population, accounting for 60–70% of all dementias. This widespread epidemic is a major concern for the aging population, with an incidence that rises sharply after 65 years of age, affecting roughly 50% of individuals aged 85 and older [1]. AD and associated dementia are estimated to afflict 50 million people worldwide, a number projected to triple by 2050 [2]. As the sixth-greatest cause of mortality worldwide, AD carries a cost of more than one trillion dollars each year [3]. Thus, the prevalence and worldwide impact establish this neurodegenerative condition as a public health and research priority. While currently there is no cure, with early diagnosis, the patient’s lifestyle may be adjusted and the progression of the disease may be slowed [3,4].

The early stage of AD, known as preclinical AD, occurs before any clinical symptoms appear. Changes in the brain can be observed through neuroimaging or analysis of biomarkers such as amyloid and tau studies in cerebrospinal fluid (CSF). The second stage is mild cognitive impairment (MCI), in which the patient is still capable of performing daily activities but experiences mild cognitive decline. The last stage is dementia, which is characterized by a significant cognitive deterioration that adversely affects everyday activities [4].

There is no single etiology for AD, but several variables, including genetics, lifestyle, and environment, play crucial roles in its development. Most notably, apolipoprotein E allele (APOE ε4) has been recognized as a significant risk factor for developing AD [5,6]. AD is characterized classically by the creation and accumulation of aberrant amyloid-beta (Aβ) plaques and neurofibrillary tangles (NFT) of hyperphosphorylated tau protein. The characteristic deposition of (Aβ) plaques and NFT in the brain are difficult to detect in vivo.

Several theories have been suggested to explain the pathophysiology of AD. The amyloid cascade theory proposes that the brain generates Aβ when amyloid-β protein precursor (APP) is cleaved by numerous proteases, such as β- and γ-secretases, releasing Aβ fragments. The activation of astrocytes and microglia by Aβ and NFT and production of proinflammatory factors such as interleukins and nitric oxide ultimately leads to increased neuroinflammation, oxidative stress, neuronal injury, and cell death [7,8,9,10].

Currently, the diagnosis of AD is a clinical one, relying on history and neuropsychological testing. Supportive biomarker evidence (neuroimaging, serum, and CSF) of AD pathology can aid in the diagnosis. In 2018, the National Institute on Aging and Alzheimer’s Association (NIA-AA) proposed criteria to define AD by its underlying pathophysiological processes known as the A/T/N framework, i.e., amyloid, tau, and neurodegeneration [11]. An accurate clinical diagnosis of AD hinges on a detailed history, evaluation of neurocognitive deficits, and biomarkers [12,13]. PET imaging and fluid biomarkers (CSF levels of Aβ and tau) are considered the gold standard for detection of underlying pathology specific to AD [14,15,16].

However, these methods are currently invasive and/or costly, leaving them poor screening tools for the general population. Thus, AD is routinely underdiagnosed and recognized only at later, more advanced clinical stages. This delay in diagnosis has been suggested as a contributing factor in the numerous unsuccessful AD treatment trials. Although there is no known cure for AD, early diagnosis is crucial to long-term prognosis. It is well understood that earlier intervention and therapy can have significant benefits for AD patients. As a result, there is a demand for novel biomarkers for AD that are not only more practical but also more sensitive and specific.

Over the past few years, there have been substantial advances involving blood-based Alzheimer’s biomarkers. Researchers have identified classic pathophysiological hallmarks of AD, such as Aβ, NFT, and markers of neurodegeneration in serum. Results from well-defined cohorts indicate these plasma biomarkers are abnormal in synchrony with CSF biomarker values in patients with AD. Blood-based biomarkers provide a cost-effective, less invasive, and serially measurable alternative for possibly diagnosing and monitoring AD [17]. While these studies may appear promising, many prerequisites need to be met prior to clinical implementation. Serum biomarkers must be further validated in human studies and guidelines must be established for proper procurement, storage, and testing. Additionally, concentrations of Aβ and tau in the serum are much lower than those in the CSF, as these molecules do not directly enter the circulation due to the presence of the blood–brain barrier. Thus, serum levels are near the lower limits of detection of many current assays, consequently losing their sensitivity for detecting narrow differences between biological samples [18]. Further trials need to define criteria for when and how these biomarkers should be utilized [19].

Although AD has been historically perceived as a brain disorder, recent studies indicate that AD also manifests in the eye, with mounting evidence of biomarkers in the retina, vitreous, cornea, and numerous other ocular structures [5,6,7] (Figure 1). Due to the current lack of accessible screening techniques for AD, alternative screening biomarkers have been a great source of recent investigation [20]. This article is a review of the latest developments of ocular biomarkers that may become a more practical tool for early diagnosis, prognostic assessment, and management response for AD.

## 2. Ocular Biomarkers in Alzheimer’s Disease

### 2.1. Retina and Optic Nerve

The retina is a delicate neurosensory tissue that lines the interior surface of the back of the eye. Retinal tissue converts light signals into nerve impulses that are transmitted to the brain through the optic nerve. The retina is comprised of nine layers: the photoreceptors (rods and cones), the external limiting membrane, the outer nuclear layer, the outer plexiform layer, the inner nuclear layer, the inner plexiform layer, the ganglion cell layer, the nerve fiber layer, and the internal limiting membrane [21]. (Figure 2) After light contacts the rod and cone photoreceptor cells, action potentials are sent to the brain via the optic nerve [21]. Thus, the neurosensory retina is a central nervous system (CNS) extension that is often regarded as a direct extension to the brain.

This neurosensory tissue not only shares an embryonic origin with the brain but exhibits similarities in anatomy, function, response to damage, and immunology. Retinal vascularization, self-regulating blood flow, and blood–tissue barrier activities are also akin to the brain. Moreover, many neurodegenerative conditions such as Alzheimer’s disease will similarly affect retinal tissue. Particularly, the hallmark pathological signs of AD, amyloid β-protein (Aβ), and neurofibrillary tangles (NFTs) of hyperphosphorylated (p)Tau protein have been identified in the retina [6,8].

Early histopathologic analyses from the late 20th century revealed reduced peripheral retinal ganglion cell layers in cadaveric AD eyes compared to control eyes. Moreover, extensive neuronal loss was noted throughout the entire retina in AD compared to control eyes, with the overall neuronal loss measured at 36.4% [22,23,24].

Contemporary studies by Lee et al. found that levels of Aβ retinal deposits were significantly higher in neuropathologically confirmed Alzheimer’s disease patients than controls. Further regional analysis demonstrated mid-peripheral Aβ levels were greater than central retina in both AD and control eyes [25]. Aβ is an appealing biomarker for the early detection of AD, as it may accumulate up to 20 years prior to clinical presentation of dementia [26].

More recently, Xu et al. confirmed higher levels of Aβ in the mid-peripheral postmortem human retinae of AD eyes compared to controls. This group further analyzed microglial and macroglial metabolic activity in these AD eyes. The connection of Aβ with glial cells was explored, as glial cells are believed to play important roles in homeostasis and clearance of Aβ in the AD retina. Normally, glial cells live in a symbiotic relationship with retinal neurons as they function to support neuronal metabolic activity. Thus, pathological insults induce microglial activation and microgliosis in neurodegenerative disease. The authors found that higher levels of Aβ in AD were associated with increased levels of microgliosis and Müller cell degeneration in AD eyes compared to controls. Xu believed this increased microglial activation in AD is a result of proinflammatory states and Aβ deposition, further leading to neurotoxicity. These results suggest that dysfunction of the Müller and microglial cells may be key features of the AD retina, thus raising the possibility of using microgliosis as a new potential biomarker for early AD detection [27].

While the presence of retinal amyloid plaques and tissue loss has long been established in AD, histopathologic evaluation is limited to postmortem analysis. The feasibility of utilizing retinal tissue as a surrogate measure of AD may be greater through more noninvasive retinal imaging techniques. Current noninvasive imaging techniques include high-resolution optical coherence tomography (OCT) and OCT angiography (OCTA). With the advent of this newer technology, our ability to identify pathology in the retinal layers and vascular system has greatly improved [28].

#### 2.1.1. Optical Coherence Tomography (OCT)

Recently, there has been an increased focus on utilizing retinal imaging as an alternative to assess for manifestations of AD. Examination of the retina has attracted significant interest, as it is more accessible and cost-effective than neuroimaging of the central nervous system (CNS). The retina is the only central nervous system tissue that is not protected by bone, thus enabling unique noninvasive imaging that offers an extraordinary window into the brain. Retinal imaging has advanced quickly in recent decades, making it possible to evaluate the retinal vasculature and neural structure easily and noninvasively. Due to the inherent challenges in detecting biomarkers in Alzheimer’s disease, retinal imaging such as optical coherence tomography (OCT) and OCT angiography (OCTA) has the potential to provide practical and efficient markers with potential diagnostic and monitoring capabilities for AD and dementia [29,30].

OCT uses light waves to obtain high resolution cross-section images of the aforementioned distinct retinal layers. OCT has evolved as a popular in vivo method for assessing retinal abnormalities, delivering high-resolution two-dimensional cross-sectional pictures and three-dimensional volumetric images of the retina [31]. While OCT is a routine part of ophthalmologic examination, recent studies have demonstrated the potential utility of OCT imaging to diagnose and monitor Alzheimer’s disease. It has been increasingly evident that retinal tissue density throughout multiple layers is reduced in Alzheimer’s disease. Numerous studies have demonstrated that individuals with AD possess considerably thinner retinas than both healthy controls and individuals with mild cognitive impairment [32,33,34]. The literature consistently describes thinning of the peripapillary retinal nerve fiber layer (RNFL) and the ganglion cell–inner plexiform layer complex (GC-IPL), along with a reduction in total macular volume in patients with AD [35,36,37,38,39,40,41,42,43]. A 2017 meta-analysis by den Haan et al. reviewed 25 studies analyzing retinal thickness in AD, mild cognitive impairment (MCI) patients and healthy controls (HCs). They noted a statistically significant reduction in mean peripapillary RNFL layer thickness and macular thickness in patients with MCI and AD [44]. Whereas mean RNFL (mRNFL) thickness tends to be decreased overall in AD patients, investigations have revealed that this thinning is frequently localized to the superior quadrants of the peripapillary RNFL [41,45,46,47,48,49]. However, a recent 2022 study by Kim et al. noted that macular RNFL thinning may even precede peripapillary RNFL thinning and may be a prognostic biomarker of cognitive decline in older individuals. The authors followed patients aged > 60 and found a strong association between decreased cognitive function test scores and thinner macular retinal nerve fiber layer [50]. Moreover, Lopez-Cuenca et al. examined the macular retinal thickness and peripapillary RNFL in healthy cognitive subjects at high genetic risk of developing AD, comparing them to age-matched controls. These high-risk patients possessed a family history of Alzheimer diseases and were ApoE ε4 carriers. They noted statistically significant reductions in the RNFL, outer plexiform layer (OPL), inner nuclear layer (INL), and inner plexiform layer (IPL) in high-risk AD patients [51].

Another prospective biomarker for the early detection of Alzheimer’s disease is choroidal thickness [52]. The choroid is a vascular tissue layer that lines the outer surface of the retina. Choroidal thickness in AD eyes has been reported to be considerably thinner in AD eyes than control eyes [53]. Alzheimer’s Disease Assessment Scale cognitive subscale (ADAS-Cog) scores and age were found to be independently correlated with choroidal thickness in a multivariable regression analysis, and subfoveal choroidal thickness was found to be a strong predictor of Alzheimer’s disease [54]. This choroidal thinning is believed to be caused by local Aβ deposition-induced vasculopathy. Studies of pathologic brain changes in AD suggest that Aβ buildup in the choroid may trigger inflammatory responses and complement activation, leading to neurodegeneration and choroidal vascular regression [42,53,55,56] (Figure 3).

#### 2.1.2. Optical Coherence Tomography Angiography (OCTA)

Optical coherence tomography systems can now obtain a retinal angiogram using a noninvasive approach known as OCT angiography (OCTA), which can offer depth-resolved volumetric details on the choroidal and retinal microcirculation by detecting the movement of flowing red blood cells and reconstructing them in structural pictures using sequential OCT scans. Given the similarity and continuity of cerebral and retinal vasculature, OCTA could be used to identify microvascular alterations in Alzheimer’s disease through retinal blood flow [57,58]. These alterations are thought to be the result of beta-amyloid plaque formation in the retina. Amyloid plaques may put pressure on retinal layers and vasculature, producing reduced blood flow, lower oxygen levels, and a lack of glucose and other nutrients. As a result, the hypoxic retina reacts by generating VEGF to encourage angiogenesis and reestablish its blood supply. Nonetheless, β-amyloid plaques restrict angiogenesis by functioning as a mechanical barrier to VEGF release, trapping it inside the plaques and blocking VEGF from reaching surrounding healthy retina [59]. In 2018, O’Bryhim et al. noted that Aβ-positive preclinical patients had a greater foveal avascular zone (FAZ) on their OCTA in comparison to healthy people [60]. Subsequent studies by Zhang in 2022 demonstrated FAZ changes were also related to medial temporal lobe atrophy (MTA) scores in individuals with early-onset dementia (EOD), whereas retinal microvasculature was correlated with white-matter hyperintensity [61]. Additionally, patients with AD and cognitive impairment have been found to have lower retinal capillary vessel density as well as flow density compared to controls [62,63,64,65,66,67,68] (Figure 4).

While promising, OCT and OCTA possess limitations in regard to detecting AD. High-quality images may become progressively difficult to obtain in patients with severe dementias, as these patients may be unable to follow commands or stay properly positioned for testing. Coexisting ocular and systemic diseases may confound the ability to assess for AD-related neurodegeneration. Ocular conditions such as age-related macular degeneration and glaucoma are common among the elderly. Additionally, systemic diseases such as diabetes are often accompanied with anomalies in the macula, optic nerve and retinal nerve fiber layer, which may interfere with the accuracy of OCT findings [69]. For this reason, most studies on OCT for AD diagnosis exclude those with eye disease, a key demographic representing an at-risk population for AD.

As reviewed above, initial research appears promising for the potential role of noninvasive retinal imaging in the screening of Alzheimer’s disease. OCT and OCT-A offer uniquely accessible and cost-effective images that have the potential to be incorporated into general screening of AD. While there have been additional confounding and conflicting studies, further research needs to be performed to replicate and clarify the role of retinal imaging techniques to assess AD biomarkers.

### 2.2. Vitreous Humor

The vitreous humor is a nonreproducible, transparent gelatinous substance that fills the posterior segment of the eye. It plays a crucial role in stabilizing the lens and contributing shape and structure to the posterior portion of the eye. The vitreous humor is composed of 98% water, with the remaining substances including collagen, hyaluronan, glycosaminoglycans, and numerous additional proteins. Various components are believed to diffuse into the vitreous via the vitreous–retinal interface. Therefore, direct sampling of vitreous fluid may provide an accessible and cost-effective means of diagnosis and prognostication for CNS pathologies. Proteomic research of vitreous humor has yielded promising results for biomarkers that could be used to identify or track the course of numerous neurodegenerative illnesses, such as Alzheimer’s [70,71]. In both human and animal research, significant amounts of Aβ40 and Aβ42 were found in vitreous humors [72,73]. Low levels of vitreous Aβ40, Aβ42, and t-tau were shown to be strongly associated with poorer MMSE scores in Wright et al.’s investigation [71]. Moreover, our group recently identified vitreous concentrations of neurofilament light chain (NfL), a known neurodegenerative biomarker in the CSF and blood, where levels of NfL were quantified in 77 vitreous samples and were significantly associated with increased levels of Aβ40, Aβ42, and t-tau [74]. Vitreous sampling can be obtained in the clinic via vitreous biopsy aspiration or in the operating room through surgical vitrectomy. Both methods of vitreous sampling are invasive and carry risks, which include retinal tears, retinal detachment, choroidal hemorrhage, vitreous hemorrhage, endophthalmitis, exacerbation of the underlying inflammatory disease, and proliferative vitreoretinopathy [75]. At this time, sampling of vitreous fluid should be reserved for patients already undergoing therapeutic vitrectomy for other ocular pathologies. While there may be a future role for vitreous sampling, further studies are needed to validate the role of vitreous biomarkers in the diagnosis and management of Alzheimer’s disease before regularly sampling vitreous tissue in the general population.

### 2.3. Aqueous Humor

The aqueous humor is a transparent, reproducible fluid generated by the ciliary body, which fills the anterior segment of the eye. It is essential for regulating intraocular pressure as well as providing nutrition and immunologic materials for the anterior structures of the eye. Protein concentrations in aqueous solutions range between 120 and 500 ng/µL. Investigative studies into aqueous composition in AD have interested many groups, as the blood–aqueous barrier is somewhat akin to the blood–CSF barriers, producing many common features between aqueous solutions [76]. The previously discussed Goldstein et al. group initially identified amyloid proteins Aβ40 and Aβ42 in aqueous humor at concentrations comparable to cerebrospinal fluid [77] Moreover, studies of animal models have shown that when Aβ42 is injected into the CSF of transgenic mouse models of AD, it promptly migrates to the aqueous humor [72,78]. Obtaining aqueous humor samples is an invasive process, similar to sampling vitreous humor, requiring the insertion of a small-gauge needle into the anterior chamber. Rare but serious risks, such as endophthalmitis and corneal abscess, are present, and there is always a danger of iris and lens trauma [79]. However, the procedure is relatively quick and straightforward, and could be a potential source of testing if future studies can consistently identify AD biomarkers in aqueous humor.

### 2.4. Lens

The crystalline lens is an epithelium differentiated organ with a biconvex shape that is located in the posterior chamber of the eye. The human lens has one of the highest concentrations of protein of any tissue, thus facilitating accommodation and adjustments in optical power [80,81]. Although the lens carries protective mechanisms, these processes slowly deteriorate with age and the lens begins to accumulate modified protein. As lens proteins gradually aggregate, a cataract begins to form, resulting in a visual opacity and the degradation of optical quality [82]. Although cataract formation is a routine aging phenomenon, certain neurodegenerative disease processes are known to accelerate the process and form distinctive opacities. The combination of the lens’s optical accessibility and high protein concentration makes it an exceptional tool for in vivo optical research on protein aggregation.

Investigations have been performed on the presence of Aβ aggregation resulting in cataract formation [83]. Studies involving the use of customized fluorescent drops that bind Aβ proteins in the eye have been employed to identify whether an individual’s cataract is Alzheimer’s-related [84]. In 2013, Kerbage et al. made major progress by identifying evidence of AD through examination of the in vivo human lens. The authors administered a topical compound that binds to aggregate Aβ peptide in AD and controls. With the use of fluorescent ligand eye-scanning techniques (FLESS), a modality capable of detecting Aβ in the crystalline lens, they were able to detect an increase in fluorescent signatures in the AD patients compared to controls. Subsequent work by Kerbage et al. in 2015 utilizing the FLESS system noted high specificity (95%) and sensitivity (85%) in detecting AD patients [84,85].

Research by Goldstein et al. showed that amyloid-β protein precursor (APP) and Aβ deposits were detected in cadaveric human lens samples of people with diagnosed AD in whom Aβ had been confirmed by autopsy. The authors also reported a cortical cataract opacification that was seen exclusively in AD patients. These supranuclear cataracts colocalised with enhanced Aβ immunoreactivity and birefringent Congo red staining. Additionally, Aβ40 and Aβ42 also were found in the cytoplasm of cortical lens fiber cells from Alzheimer’s disease patients, indicating that Aβ may cause region-specific lens protein accumulation and supranuclear cataracts [86].

Moncaster et al. further expanded on this work, identifying this cortical lens opacity in individuals affected by trisomy 21—Down syndrome (DS). Chromosome 21 contains the gene encoding for AβPP, and thus carrying an extra copy of the gene predisposes DS patients to accumulate amyloid deposition within the brain. DS patients often develop cognitive impairments and early dementia similar to AD patients [87]. Peptide sequencing, immunoblot analyses and ELISA studies were subsequently performed to confirm the presence of human Aβ isoforms (Aβ40 and Aβ42). Total Aβ levels were consistently higher in lens and brain tissue of Down syndrome patients relative to normal controls and equivalent to the values previously reported in AD. These findings demonstrated that the lens experiences the same classical amyloid pathology as the brain. Additionally, increased Aβ accumulation is a critical pathogenic factor that links the crystalline lens and brain in both AD and Down syndrome [86,87].

The crystalline lens has received considerable attention, but further research is required to confirm the presence of AD biomarkers, particularly in preclinical AD. Utilizing lens biomarkers has the potential to noninvasively facilitate diagnosis and monitor progression, as cataract lens extraction remains one of the most common procedures worldwide. Considering the unique access to lens tissue through both cataract surgery and through noninvasive in vivo examination, the lens remains an intriguing potential source for AD diagnosis. However, many recent studies evaluating the presence of AD-related proteins have been unable to replicate these findings. Various groups have since analyzed postmortem crystalline lenses in AD patients and have not found the presence of neuropathologic changes such as Ab aggregates or supranuclear opacities. These inconsistencies may be due to multiple factors, including discrepancies in laboratory techniques and differing diagnostic criteria for AD study patients [88,89,90,91,92]. Additional research in preclinical Alzheimer’s disease patients is required to confirm if abnormal protein accumulation in crystalline lenses could be utilized as a predictive tool.

### 2.5. Pupil

The pupil is the central opening of the iris that allows light to enter the eye so it can be focused on the retina. Pupillary activity occurs through divisions of the autonomic nervous system. The central pupillary zone and the peripheral ciliary zone are concentric zones on the front surface of the iris. The pupillary sphincter, which is responsible for pupil constriction, is a 1 mm muscle that concentrically encircles the pupil. Physiologic pupil constriction is regulated by the parasympathetic nervous system’s cholinergic innervation of the pupillary sphincter and occurs as a result of light stimulation or focus on an object at close distance. Catecholaminergic innervation of the pupillary dilator by the sympathetic nervous system mediates physiological dilation of the pupil [93]. The pupil light reflex (PLR) is the pupil’s reaction to a strong flash of light, which involves rapid constriction followed by dilatation back to normal size. Because the pupil light reflex is primarily a parasympathetic cholinergic response, it may be disrupted if central cholinergic reduction in AD spreads to the parasympathetic oculomotor system [20]. Further, pathologic changes in cerebral structures and autonomic nervous system governing the pupillary light reflex may arise in Alzheimer’s disease.

Increased pupillary dilation during cognitive task performance is a proven, objective psychophysiological marker of the brain’s cognitive resource allocation, also known as cognitive effort. Haj et al. demonstrated that more complex cognitive tasks were reflected in a larger pupil size in both AD and healthy older persons [77]. In their investigation, Granholm et al. measured pupillary responses during digit span recall in 918 individuals whose ages ranged from 56 to 66. At lower cognitive demands, they discovered that amnestic mildly cognitively impaired subjects had more pupil dilation than cognitively normal adults [94]. In similar research, Kremen et al. assumed that even cognitively normal middle-aged adults with higher hereditary risk of AD (based on Alzheimer’s disease polygenic risk scores) would have atypical pupillary responses. They subsequently demonstrated that these higher-risk individuals had significantly greater pupil dilation during high cognitive demand [95]. However, the effect of Alzheimer’s disease on pupillary diameter at rest has been controversial in several studies, most likely due to methodological variations, such as measurement settings and sample numbers. Kawasaki and colleagues looked at 16 AD patients and 16 controls, and they alternately found that the baseline pupil size was considerably smaller in AD patients. They also reported that the pupillary contraction amplitude to all red and blue lights was lower in AD patients; however, this difference did not achieve statistical significance [96]. Additional studies with limited sample sizes (ranging from 9 to 24 AD patients) further showed lower baseline pupillary diameters in AD compared to healthy participants; moreover, other research found no difference in baseline pupil diameters between AD, MCI, and controls [97,98,99].

Overall, further studies must be performed to assess the role of pupil evaluation in detecting cerebral AD pathology. However, due to its relative ease of access and noninvasive examination, pupillary characteristics may be a worthwhile target of further research for the diagnosis of Alzheimer’s disease and surveillance of cognitive impairment.

### 2.6. Cornea

The cornea is an avascular and transparent connective tissue that works as the eye’s principal infectious and structural barrier. It has five distinct layers, three cellular (epithelium, stroma, and endothelium), and two interfaces (Bowman membrane and Descemet membrane). The cornea and tear film work in tandem to provide the eye with an appropriate anterior refractive surface [100]. The cornea provides 40–44 D of refractive power and is responsible for about 70% of the total refraction. The anterior stroma, sub-basal plexus, and epithelium of the cornea all contain dense concentrations of nerve fibers that do not obstruct light transmission, but instead sustain the cornea’s abundant nerve supply [101]. Several illnesses, including AD, result in negative ocular surface changes that may be related to the unfavorable consequences of corneal nerve dysfunction. It has been shown that corneal nerve degeneration occurs in AD by directly examining the morphology of corneal nerves [102,103]. The usage of corneal confocal microscopy (CCM) for quick, noninvasive clinical evaluation of corneal nerves has increased significantly, particularly in recent years. Ponirakis et al. investigated the relationship between corneal nerve fiber measurements, cognitive function and functional independence in individuals with MCI and dementia. When individuals with MCI and dementia were compared to healthy controls, they discovered a gradual decrease in corneal nerve fiber density (CNFD), branch density (CNBD), and fiber length (CNFL). They also demonstrated that, after controlling for confounders, all three corneal nerve fiber measurements were strongly related to cognitive function and functional independence in MCI and dementia [104]. This team also carried out a further investigation in 2021 to examine the relationship between cerebral ischemia and the loss of the corneal nerve and brain atrophy in MCI and dementia. They discovered that participants with both subcortical and cortical ischemia had lower hippocampal volume, shorter corneal nerve fiber length, and larger ventricular volume than those with only subcortical ischemia or no ischemia [105]. More recently, Ponirakis assessed the relationship between corneal nerve morphometry and MRI brain volumetry and cognitive performance in MCI and dementia. After adjusting for potential confounders, he discovered cognitive impairment was related to corneal nerve fiber density, length, branch density, whole brain, hippocampus, cortical gray matter, thalamus, amygdala, and ventricle volumes [106].

Similar research by Al-Janahi et al. on 182 individuals with no cognitive impairment (NCI), MCI, and dementia, including AD, revealed that individuals with MCI and dementia had progressively smaller corneal nerve fiber density and fiber length than those with NCI [107]. Ornek’s research on assessing corneal sensitivity in neurodegenerative diseases and comparing it to age- and gender-matched controls found that mean corneal sensitivity was considerably lower in AD when compared to controls [108].

Dutescu et al. discovered amyloid precursor protein (APP) in the corneal epithelium of transgenic animal models of Alzheimer’s disease [109]. In a different study, Choi et al. investigated the metabolism of APP, the expression of AD-related genes and proteins, and the expression of Aβ degradation enzymes in corneal fibroblasts to determine whether the cornea can be used to identify possible biomarkers for the early diagnosis and development of AD. They discovered that APP and proteins involved in APP processing and Aβ degradation were expressed by corneal fibroblasts and the corneal epithelium in this investigation [110]. However, while the limited available studies appear promising, more comprehensive study in this field is needed to reach more clear conclusions on the value of the corneal tissue as a source of biomarkers.

### 2.7. Tear Fluid

Tear fluid is composed of an aqueous–lipidic layer that contains proteins, mucins, lipids, water, and electrolytes. The tear film functions to lubricate and nourish the eyelids, conjunctiva, and cornea. It works to maintain a smooth surface for light refraction and provides the cornea with nutrition. It allows white blood cells to reach the conjunctiva and cornea, facilitating immune defense, using both particular and general antibacterial agents to protect the eye surface against infections [111]. Tears, an easily accessed body fluid with a generally simple and stable composition, are an obvious research focus for neurodegenerative disorders biomarkers such as Alzheimer’s disease [112]. In 2016, Kalló et al. examined tear samples from 14 AD patients and 9 age-matched controls and discovered that the flow rate, which was 6 ± 2 µL/min in controls, had significantly increased to 12 ± 2 µL/min in AD patients. Additionally, the concentration of tear protein had increased significantly from 4.4 ± 1.4 µg/µL in controls to 8.8 ± 2.9 µg/µL in AD patients. Furthermore, the tear fluid of Alzheimer’s disease patients included considerably lower amounts of lysozyme, lipocalin 1, and lacritin, as well as higher levels of dermcidin. A combination of four tear proteins—lipocalin 1, dermcidin, lysozyme C, and lacritin—was shown to have 81% sensitivity and 77% specificity for Alzheimer’s disease [113].

Del Prete et al. discovered Aβ42 using an immunocytochemistry assay in the tear smears of two individuals with a family history of AD, while also demonstrating its absence in the tears of a healthy individual [114]. Gijs et al. conducted a more detailed study of classical AD biomarkers in the tears of patients with various degrees of cognitive impairment (subjective cognitive decline-SCD, mild cognitive impairment-MCI, and Alzheimer’s dementia). They discovered that the great majority (>94%) of tear fluid samples had quantities of Aβ40 and t-tau that could be detected. When compared to patients without neurodegeneration, patients with neurodegeneration had higher amounts of t-tau in their tears. CSF Aβ42 and tear t-tau were shown to be negatively correlated [115]. Additionally, Kenny and colleagues evaluated tear protein and microRNA content in samples from patients with MCI and AD. They discovered elongation initiation factor 4E (eIF4E) as a novel protein found solely in Alzheimer’s disease samples. Moreover, total microRNA abundance was discovered to be greater in tears from Alzheimer’s patients. MicroRNA-200b-5p was found as a possible biomarker for Alzheimer’s disease, with increased levels detected in AD tear fluid samples compared to controls [116]. A more clinically practical correlation was also noted by Ornek et al. His group identified shorter tear breakup times (TBUT) and lower Schirmer test scores, both regularly performed clinic tests, to be characteristics of Alzheimer disease [108].

However, the practical incorporation of tear fluid analysis for AD is still distant. Further large-scale studies must be performed to establish consistent biomarker interpretations and reliable testing protocols. Additionally, current immunoassays for proteomic studies often require large sample volumes. Tear fluid collection, although noninvasive, is relatively limited and increasingly diminished in the elderly demographic of AD. It is well known that tear function declines with age, perhaps requiring elderly patients to return on multiple visits to obtain an adequate sample [117]. Moreover, tear fluid, in contrast to CSF and blood, currently requires sample preprocessing that involves dilution, further reducing protein concentration [115]. Overall, due to the simple sample collection methods and availability, tear analysis appears to be a worthwhile target of further investigation as a screening tool for AD [118].

## 3. Conclusions

Alzheimer’s disease is the most common type of dementia and can be defined as a slowly progressive neurodegenerative disease characterized by amyloid plaques and neurofibrillary tangles. AD follows a gradual continuum that begins from an asymptomatic stage with biomarker evidence of AD (preclinical AD) through mild cognitive impairment to, ultimately, AD dementia. While there is currently no cure for AD, early diagnosis and management can significantly improve patient morbidity and life expectancy. Currently, the diagnosis of Alzheimer’s disease is a clinical one, often supplemented by biomarker evidence of pathology using the ATN classification system (amyloid, tau, neurodegeneration). Current testing such as neuroimaging and cerebrospinal fluid biomarkers are invasive, expensive, or not widely available, leaving them poor screening tools for the general population. Thus, AD is routinely underdiagnosed and recognized only at later, more advanced clinical stages.

The eye, an extension of the central nervous system, has long been studied for insight into neurodegenerative conditions. Over the last decade, significant advancements have been made in our understanding of AD and the role of ocular tissue as a potential biomarker (Table 1). Ocular biomarkers hold the potential to provide objective, affordable, and easily accessible markers with potential diagnostic and monitoring capabilities. Ocular imaging devices such as optical coherence tomography (OCT) and OCT angiography additionally provide quick noninvasive images that could reflect the underlying pathology.

This review summarizes current research for detecting biomarkers of Alzheimer’s disease in ocular tissue. Future studies are needed to investigate the potential utility of ocular biomarkers as a more practical tool for early diagnosis, prognostic assessment, and management response to treatment for AD.

## Figures and Tables

**Figure 1 life-13-00726-f001:**
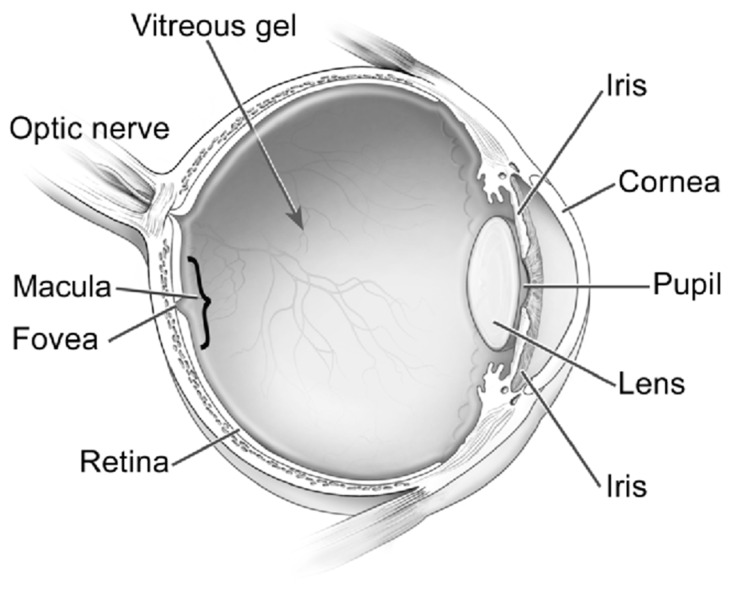
Ocular structures. Reproduced with permission from U.S. National Eye Institute Media Library.

**Figure 2 life-13-00726-f002:**
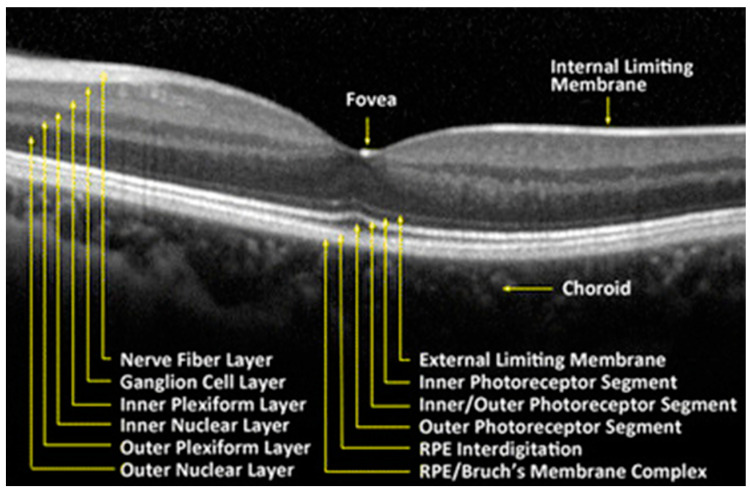
Retinal tissue layers. Reproduced with permission from BioMedical Engineering Online; Springer Nature, [July 2016].

**Figure 3 life-13-00726-f003:**
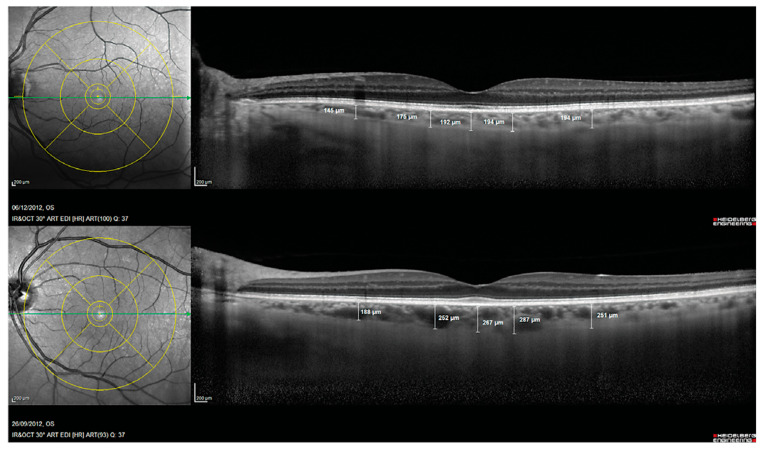
Choroidal thickness measurements assessed with spectral domain optical coherence tomography in a patient with Alzheimer’s disease (**Top**) and in a control subject (**Bottom**). Reprinted from [53], Copyright 2014, with permission from IOS Press.

**Figure 4 life-13-00726-f004:**
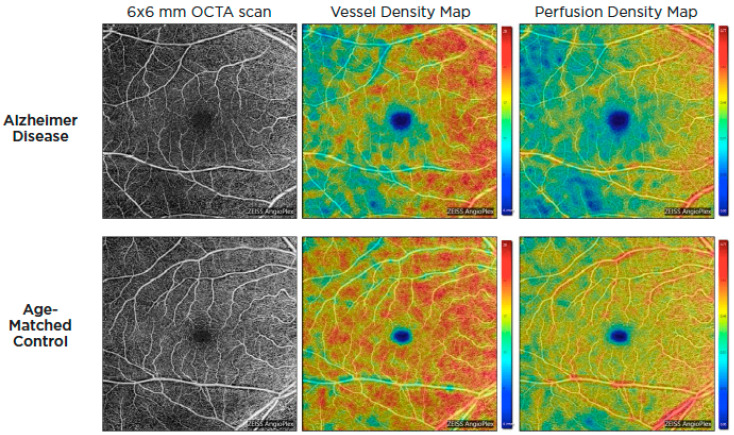
OCTA images of the superficial capillary plexus of the right eye from a patient with AD (**top row**) and an age-matched control (**bottom row**). Corresponding quantitative color maps of vessel density (**middle column**) and perfusion density (**right column**) with their respective color scales on the right show decreased vessel density and perfusion density in the subject with AD compared to the control. Reproduced with permission from American Academy of Ophthalmology *EyeNet* magazine; [July 2019].

**Table 1 life-13-00726-t001:** Summary of ocular biomarker findings and limitations.

Ocular Biomarker	Significant Findings	Limitations
Retinal tissue	Increased levels of Aβ retinal deposits Reduced peripheral retinal ganglion cell layers and overall increased neuronal tissue loss Increased microglial activation and Müller cell degeneration	Evaluation of histopathology in AD is limited to postmortem analysis
Optical Coherence Tomography (OCT)	Noninvasive and cost-effective imaging technique Thinning of retinal tissue density—particularly the RNFL, the GC-IPL, choroidal thickness, and reduction in total macular volume	Coexisting ocular and systemic diseases can affect the accuracy of OCT results Requires prolonged positioning and following commands that may be increasingly difficult in patients with advanced dementia or concurrent motor dysfunction
Optical Coherence Tomography Angiography (OCTA)	Noninvasive and cost-effective, high-resolution angiography Identify microvascular alterations in Alzheimer’s disease through retinal blood flow Larger foveal avascular zone (FAZ) Reduced retinal capillary vessel density and flow density	
Vitreous Humor	Accessible and cost-effective technique Presence of Aβ40, Aβ42, t-tau, and NFL in vitreous samples	Vitreous sampling is invasive and the procedure carries risks Should be reserved for patients undergoing therapeutic vitrectomy for concurrent eye conditions
Aqueous Humor	Presence of Aβ40, Aβ42 in aqueous humor at concentrations comparable to CSF levels	Aqueous sampling is invasive and the procedure carries risks
Lens	Presence of amyloid-β protein precursor (APP) and Aβ deposits in lens in AD patients Detection of Aβ aggregation in the lens using in vivo scanning techniques Easy access to lens tissue through both cataract surgery and noninvasive in vivo examination	Currently inconsistent results in confirming AD-related proteins in the lens or AD-related supra-nuclear opacities across various studies
Pupil	Easy access and noninvasive examination Increased pupillary dilation during higher-demand cognitive tasks	Currently inconsistent results demonstrating the effect of AD on pupillary diameter
Cornea	Noninvasive clinical evaluation of corneal nerves Gradual decrease in corneal nerve fiber density (CNFD), branch density (CNBD), and fiber length (CNFL) Lower mean corneal sensitivity Presence of APP in the corneal fibroblasts and epithelium	Additional studies required to validate results
Tear Fluid	Easily accessible body fluid with noninvasive acquisition Increased tear flow rate and tear protein concentration Lower amounts of lysozyme, lipocalin 1, and lacritin Presence of Aβ40, t-tau, microRNA and elongation initiation factor (4EeIF4E) Shorter tear breakup time (TBUT) and lower Schirmer test scores	Limited and diminished tear production in the elderly population Reduction in protein concentration for testing due to sample preprocessing

## Data Availability

No new data were created or analyzed in this study. Data sharing is not applicable to this article.

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
