# Peer review of "The Eye as a Diagnostic Tool for Alzheimer’s Disease"

_life, 2023, doi:10.3390/life13030726_

Round 1
Reviewer 1 Report
The work is well written and it's a good summer of all technics to detect alzheimer.
Author Response
Thank you for your review and positive comments.
Reviewer 2 Report
It is meaningful of to find the biomarkers of Alzheimer’s disease for the early diagnosis and management of the disease. This manuscript summarizes the current research for detecting biomarkers of Alzheimer's disease in ocular tissue of retina, vitreous humor, aqueous humor, lens, pupil, cornea, tear fluid. The manuscript could be published in the journal of life with minor revised.
1. The manuscript will become more readable if more figures are given in it. For example, Cross-sectional view of retina captured by OCT, imaging of retinal capillary network using OCTA.
2. The Conclusion part of the manuscript should be focused on the biomarkers of Alzheimer's disease in ocular tissue. The first two paragraphs could be summarized in one paragraph.
Author Response
Thank you very much for your review. I have now included images of OCTA and OCT with choroidal measurements, demonstrating the thinning in Alzheimer's disease. Permission has been obtained for each image and included in the citations. I have also amended the conclusion.
Reviewer 3 Report
The authors have made a consolidated review of ocular biomarkers for early diagnosis of Alzheimer’s disease.
The authors should consider having self-explanatory figures with detailed legends rather than just adapting to a figure that's already available.
I recommend the authors to produce representative OCT and OCTA images that best represent severity of AD in comparison to control subjects.
A table with significance and limitations of all ocular biomarkers would be more useful for the readers.
Author Response
Thank you very much for your review. I have now included representative images of OCTA and OCT with choroidal measurements, demonstrating the thinning in Alzheimer's disease. Permission has been obtained for each image and included in the citations. I have also included a comprehensive table summarizing the significance and limitations as you recommended.